# Moderate Aerobic Exercise Training Prevents the Augmented Hepatic Glucocorticoid Response Induced by High-Fat Diet in Mice

**DOI:** 10.3390/ijms21207582

**Published:** 2020-10-14

**Authors:** Jonatan Dassonvalle, Francisco Díaz-Castro, Camila Donoso-Barraza, Carlos Sepúlveda, Francisco Pino-de la Fuente, Pamela Pino, Alejandra Espinosa, Mario Chiong, Miguel Llanos, Rodrigo Troncoso

**Affiliations:** 1Laboratorio de Investigación en Nutrición y Actividad Física (LABINAF), Instituto de Nutrición y Tecnologia de los Alimentos (INTA), Universidad de Chile, Santiago 7830490, Chile; jonatan.dassonvalle@gmail.com (J.D.); fdiaz@inta.uchile.cl (F.D.-C.); camidonoso.17@gmail.com (C.D.-B.); csepulvedag@ug.uchile.cl (C.S.); pamela.pino@inta.uchile.cl (P.P.); 2Departamento de Tecnología Medica, Facultad de Medicina, Universidad de Chile, Santiago 8380453, Chile; pinodelafuente.francisco@gmail.com (F.P.-d.l.F.); bespinosa@med.uchile.cl (A.E.); 3Advanced Center for Chronic Diseases (ACCDiS), Facultad de Ciencias Químicas y Farmacéuticas & Facultad de Medicina, Universidad de Chile, Santiago 8380492, Chile; mchiong@ciq.uchile.cl; 4Laboratorio de Nutrición y Regulación Metabólica, INTA, Universidad de Chile, Santiago 7830490, Chile; mllanos@inta.uchile.cl

**Keywords:** glucocorticoids, FKBP51, liver, obesity, exercise

## Abstract

Glucocorticoids (GCs) are critical regulators of energy balance. Their deregulation is associated with the development of obesity and metabolic syndrome. However, it is not understood if obesity alters the tissue glucocorticoid receptor (GR) response, and moreover whether a moderate aerobic exercise prevents the alteration in GR response induced by obesity. Methods: To evaluate the GR response in obese mice, we fed C57BL6J mice with a high-fat diet (HFD) for 12 weeks. Before mice were sacrificed, we injected them with dexamethasone. To assess the exercise role in GR response, we fed mice an HFD and subjected them to moderate aerobic exercise three times a week. Results: We found that mice fed a high-fat diet for 12 weeks developed hepatic GC hypersensitivity without changes in the gastrocnemius or epididymal fat GR response. Therefore, moderate aerobic exercise improved glucose tolerance, increased the corticosterone plasma levels, and prevented hepatic GR hypersensitivity with an increase in epididymal fat GR response. Conclusion: Collectively, our results suggest that mice with HFD-induced obesity develop hepatic GR sensitivity, which could enhance the metabolic effects of HFD in the liver. Moreover, exercise was found to be a feasible non-pharmacological strategy to prevent the deregulation of GR response in obesity.

## 1. Introduction

Obesity is excess or abnormal fat accumulation that increases the risk of developing insulin resistance, type 2 diabetes, cancer, and cardiovascular disease, among others [1]. Obesity is a far more complicated phenomenon than simply the balance between energy intake and expenditure. Several endogenous and exogenous factors can favor a positive energy balance and weight gains, such as metabolic inflexibility, stress, endocrine deregulations, poor sleep, and medicines with weight gain as a side effect [2,3,4,5,6]. Thus, it implies deregulation of several biological systems involved in the energy homeostasis [7]. Among these systems, the hypothalamic–pituitary–adrenal axis (HPA), with its end-product, glucocorticoids (GCs), is a major player due to their influence in energy homeostasis [8].

GCs’ actions are mediated by a ubiquitous intracellular protein, the glucocorticoid receptor (GR), which belongs to a large family of transcription factors known as nuclear hormone receptors. The canonical response to GCs occurs through direct regulation of gene transactivation or transrepression [9]. GCs are critical regulators of energy balance; part of their actions on metabolism are through catabolic activity. GCs cause the mobilization of energetic substrates from peripheral sources such as the liver, skeletal muscle, and adipose tissue [10]. Among the genes regulated by GCs, FK506-binding protein 51 (FKBP51) has been described as an important marker for GR sensitivity and bioavailability [11,12]. FKBP51 is a co-chaperone that modulates GC responses by negatively regulating GR activity, with their role in inflammation, autophagy, and insulin resistance having recently been discovered [13]. Pyruvate dehydrogenase kinase 4 (PDK4) is a mitochondrial isoenzyme that regulates glucose oxidation to acetyl-Coenzyme A (acetyl-CoA) [14] and participates in the development of several diseases [15]. The Krüppel-like factor 15 (KLF15) is a transcription factor that controls the main pathways of amino acid, glucose, and lipid metabolism, and mediates the ergogenic effects of GCs [16].

Deregulation of GC homeostasis is associated with the development of obesity and the alteration of glucose and lipid metabolisms [17]. Moreover, prolonged treatment or high doses of GCs lead to undesirable effects such as weight gain, hyperglycemia, and abdominal obesity [18]. However, research on GCs and obesity interactions has focused primarily on the role of GCs in triggering obesity and metabolic syndrome. While these studies have been important in terms of understanding the role of GC-induced obesity, whether the GCs’ response is altered once obesity has already been established requires clarification. Few studies have evaluated the effects of GCs in a diet-induced obesity model. Shpilberg et al. (2012), in a type 2 diabetes rapid onset rodent model, showed that the combination of exogenous GCs with a high-fat diet (HFD) rapidly induced insulin resistance, severe hyperglycemia, and fat deposition when compared to GCs or HFD alone [19]. Interestingly, chicks fed for 6 days with an HFD that were then sacrificed 3 h after intracerebroventricular injection with dexamethasone had reduced expression of appetite control-related genes in the hypothalamus in comparison with chicks fed with a control diet [20]. However, how obesity affects the GCs’ response in metabolic tissue is still not fully understood.

Regular physical activity (i.e., exercise) is one of the most effective non-pharmacological treatments against metabolic disorders associated with obesity and their complications. Exercise is widely used to control or treat obesity-associated diseases because of its actions on different tissues such as muscle, bone, liver, heart, vessels, and adipose tissue, among others [21]. Acute exercise also stimulates the HPA axis, increasing circulating GCs that mobilize fuel sources and moderate the immune system during the muscle damage caused by exercise [22]. On the other hand, aerobic exercise produced intracellular adaptation in a tissue-specific manner; it increased GR and the enzyme 11β-hydroxysteroid dehydrogenase type 1 (11β-HSD1) in visceral adipose tissue but reduced GR and 11β-HSD1 in skeletal muscle in Syrian hamsters [23]. Exercise training reduced hyperglycemia, hyperinsulinemia, muscle glycogen loss, and atrophy in a model of dexamethasone-treated rats [24]. Moreover, aerobic and resistance training prevented the increase in the atrophy program and reduced insulin sensitivity caused by dexamethasone treatment in rats [25,26,27]. Accordingly, exercise ameliorates the harmful effects of GCs. However, the impact of GC response in a model of HFD-induced obesity has not been explored.

In this work, we investigated the consequences of HFD-induced obesity in the GR sensitivity in relevant tissues that control energetic homeostasis and the effects of moderate aerobic training in this response. Our hypothesis was that HFD-induced obesity would deregulate GCs sensitivity, which is prevented by moderate aerobic exercise training.

## 2. Results

### 2.1. Corticosterone Plasmatic Levels Were Altered in High-Fat Diet Mice

The effect of HFD-induced obesity in GCs response was evaluated in C57BL/6J male mice fed for 12 weeks with an HFD or control diet (CD) (Figure 1A). As expected, in the first 4 weeks, the body weight of CD and HFD mice increased at the same rate. However, from week 5, HFD mice had a higher increase in their body weight compared to CD mice (Figure 1B). Moreover, increased fasting glucose level (131 ± 3.7 vs. 178 ± 5.7 mg/dL, *p* < 0.05; Figure 1C) and glucose tolerance test (GTT) were increased in the HFD group (Figure 1D,E). To assess GR sensitivity, we injected mice with the synthetic GC dexamethasone, and 8 h later they were euthanized. Plasma corticosterone levels were higher in HFD mice than CD-fed mice. Corticosterone plasmatic levels of both CD and HFD mice showed a significant decrease in dexamethasone treatment (Figure 1F). These results suggested that HFD increased corticosterone plasmatic levels without a reduction in GR sensitivity.

### 2.2. High-Fat Diet Altered GC Response in the Liver, Skeletal Muscle, and Epididymal White Adipose Tissue

To study the effect of the HFD-induced obesity in the GCs response, we assessed the dexamethasone-induced expression of FKBP51, PDK4, and KLF15 as readouts to evaluate GC response in the liver, skeletal muscle (gastrocnemius), and white adipose tissue (epididymal fat). Real-time qPCR showed a dexamethasone-dependent hyper-induction of FKBP51 and PDK4 in the liver of obese mice, with no changes in KLF15 (Figure 2A). In gastrocnemius muscle, a similar response to dexamethasone was found in FKBP51 and KLF15, but not in PDK4. In this tissue, basal PDK4 mRNA levels were increased, and no changes were observed upon dexamethasone treatment (Figure 2B). In epididymal fat, dexamethasone increased the FKBP51 mRNA level in CD and HFD mice, but only the KLF15 mRNA level in CD. No statistically significant changes in PDK4 was found. (Figure 2C). These data suggest that HFD affects GC response in a tissue-specific manner. 

To further investigate how the HFD modulates GC response, we measured GRα mRNA levels in the liver, gastrocnemius, and epididymal fat. Our result shows that dexamethasone reduced the GRα mRNA levels in the liver and gastrocnemius in both HFD and CD mice (Figure 3A,B). In epididymal fat, we found a basal increase in GRα mRNA levels in the HFD group, which was further increased by dexamethasone treatment (Figure 3C). A basal increase in 11β-HSD1 mRNA levels was found in HFD mice liver, and its levels were reduced by dexamethasone (Figure 3D). Under the same conditions, we evaluated liver protein levels of FKBP51 and glucose 6-phosphatase (G6Pase). G6Pase is the key enzyme that regulates the gluconeogenic and glycogenolytic pathways [28]. Liver FKBP51 protein levels showed the same pattern as mRNA, with the increase by dexamethasone being significantly higher in the HFD group (Figure 4E,F). HFD increased basal G6Pase protein levels, but G6Pase mRNA levels did not increase by dexamethasone treatment (Figure 4E,G). These data suggest that HFD-induced obesity triggers a GCs hypersensitivity in the liver.

### 2.3. Physical Training Increased the Corticosterone Plasmatic Levels

Having established the effect of HFD-induced obesity in GC response, we studied whether a moderate aerobic training abrogated the observed alteration. We used a similar protocol, with the addition a moderate aerobic training three times a week for 12 weeks starting at the same time as the diet (Figure 4A). Exercise did not affect the body and tissue weights but improved fasting glucose and glucose tolerance compared to the HFD group (Table 1, Figure 4B,E). Surprisingly, the corticosterone levels in CD-fed aerobic-trained mice were approximately four times higher in comparison with CD-fed sedentary mice (530 ± 76 vs. 132 ± 18, *p* < 0.05). A similar effect was observed when HFD-fed trained mice were compared with HFD-fed sedentary mice (980 ± 52 vs. 321 ± 54, *p* < 0.05). Then, we assessed the effect of dexamethasone on plasma corticosterone levels. The results showed that dexamethasone injection induced a decrease in corticosterone levels only in the HFD-fed mice (Figure 4F).

### 2.4. Moderate Aerobic Exercise Prevented the Hypersensitivity of Hepatic GC Response 

Our findings indicate that HFD promotes hepatic GCs hypersensitivity. When HFD-fed mice were subjected to physical training, this hypersensitivity, measured by the increase in FKBP51 and PDK4 mRNA levels, was abrogated (Figure 5A). On the other hand, the gastrocnemius muscle showed a differential response to exercise. The HFD group showed a higher response to dexamethasone than the CD group (Figure 5B). Interestingly, aerobic training promoted a higher increase in FKBP51 and PDK4 mRNA levels in response to dexamethasone (Figure 5C) in comparison to sedentary mice (Figure 2C). For KFL15, the HFD abrogated the increase induced by dexamethasone in the epididymal fat (Figure 5C).

Next, to evaluate the effect of moderate aerobic training in dexamethasone-induced gene expression, we compared the GR target genes FKBP51, PDK4, and KLF15 in the liver, gastrocnemius, and epididymal fat in sedentary and training mice. We observed that exercise produced a decrease in the GR response in the liver that was independent of diet. The FKBP51 and PDK4 mRNA expression were significantly lower in the CD and HFD groups than the sedentary HFD group. In the epididymal fat, exercise induced an increase in FKBP51 mRNA levels in the CD group and PDK4 mRNA levels in the HFD group. Finally, in the gastrocnemius, we did not observe any exercise effect in GCs response (Table 2).

To further investigate how moderate aerobic training alters GCs response, we measured the GRα mRNA levels in the liver, gastrocnemius, and epididymal fat. In the liver, dexamethasone reduced the GRα expression only in the CD group (Figure 6A). No significant changes were observed in the gastrocnemius (Figure 6B). In the epididymal fat, mice that exercised showed a decrease in GRα mRNA levels in response to dexamethasone, independent of diet (Figure 6C). The 11β-HSD1 mRNA levels were reduced by dexamethasone in the CD group. However, the HFD group presented lower basal levels of 11β-HSD1 that was not affected by dexamethasone (Figure 6D). The hepatic FKBP51 protein levels were increased by dexamethasone at the same extent, independent of diet (Figure 6E,F). Regarding hepatic G6Pase protein levels, HFD fed mice who exercised had higher levels of G6Pase, which was decreased upon injection with dexamethasone (Figure 6E,G). All these results suggest that moderate aerobic training modifies the GCs response in the liver and epididymal fat. Moreover, exercise could prevent hypersensitivity of hepatic GC response.

## 3. Discussion

Although the effects of an HFD on metabolic function and hormone deregulation has been extensively studied [29,30,31], the tissue responsiveness towards GCs under an HFD is far from being understood. Thus, our study constitutes new evidence in this relevant field. In the present study, we investigated the effect of HFD-induced obesity in the GC response in the liver, skeletal muscle, and adipose tissue. Our results showed that HFD-fed mice developed a hepatic sensitivity to GCs. Moreover, we showed that this alteration in GC response was prevented by moderate aerobic training.

Circulating GC levels change during the day in the circadian rhythm and in response to physiological cues and stress [9]. The deregulation in GCs action is supposed to have a key role in the development of metabolic syndrome [32]. Moreover, GR antagonism shows promising results in metabolic disease, improving weight control, glucose tolerance, and fatty liver, among others [33,34]. Several reports have shown that an HFD induced an increase in corticosterone levels [35,36,37,38,39]. Accordingly, here, we show increased corticosterone plasma levels in the HFD-fed mice over 12 weeks.

GCs have a pivotal role in the negative feedback of the HPA axis. Elevated circulating cortisol or corticosterone induces a decrease in corticotropin-releasing hormone (CRH) and adrenocorticotropic hormone (ACTH) release, decreasing the GC production by the adrenal gland [9]. However, whether HFD-induced obesity triggers deregulation in HPA axis feedback is not clear. Here, we showed that when injected with dexamethasone, HFD- and CD-fed mice had lower corticosterone levels than the vehicle. Thus, the lower corticosterone levels in dexamethasone-injected mice suggest that an HFD does not alter negative feedback in the HPA axis. This observation implies that the increase in corticosterone levels are not a secondary response due to deregulation in the negative feedback.

The GC response depends on several factors, including circulating and intracellular hormone levels, GR expression, and the ability to transduce the signal [40]. A reduced GC sensitivity or resistance has been reported in patients with chronic treatment with GCs, including those with chronic stress, elderly individuals, and individuals with cardiovascular diseases [41]. By contrast, GC hypersensitivity is a very rare state that has been described in case reports [42,43,44]. Recently, Quinn et al. showed that female ovariectomized mice develop fatty liver through a hepatic GR hypersensitivity [45]. Here, we show that feeding for 12 weeks with HFD induced an increase in the hepatic GCs’ sensitivity without affecting GC response in gastrocnemius or epididymal fat. We measured the GR transcriptional activity by expressing their target genes FKBP51, PDK4, and KLF15. In the liver, FKBP51 and PDK4 showed a higher increase in response to dexamethasone in the HFD-fed mice compared with CD-fed mice, with the exercise being able to prevent this increase. Regarding the expression of KLF15, a significant increase in gastrocnemius but not in the liver and adipose tissue was observed. These results agree with the literature, showing that no significant over-expression of KLF15 in the liver was observed in mice stimulated with a pulse of dexamethasone 6 h before sampling. Moreover, these animals displayed a significant increase of KFL15 at the muscular level, as did our results [46]. However, in vitro studies indicate that the co-stimulation with dexamethasone/8-(4-Chlorophenylthio)adenosine 3′,5′-cyclic monophosphate (8-CPT-cAMP) increases the KLF15 expression in rat primary hepatocytes, which could have an important role in gluconeogenesis [47]. In the gastrocnemius, the HFD did not affect the mRNA expression of FKBP51 and KLF15. However, it increased the mRNA expression of PDK4 compared to the control PBS group, concordant with the literature, which showed a PDK4 increase on skeletal muscle of obese mice [48]. However, moderate aerobic training did not affect the gastrocnemius GC response.

Consistent with mRNA liver expression, we observed an increase in the FKBP51 protein levels. Moreover, we measured G6Pase protein levels that were affected by the diet, which was in accordance with previous reports [49,50,51]. However, in contrast to what we expected, dexamethasone did not increase their levels. The GP6ase gene expression is regulated by hormones that include insulin, glucagon, and GCs. However, the promoter contains positive and negative GR-responsible elements, suggesting a tightly coordinated response [28].

FKBP51 is known for its essential regulation of GR activity. It forms a heterocomplex with HSP90 and GR, which regulates the folding and trafficking of the receptor [52]. FKBP51 can reduce GR binding to GCs and delay the GR translocation to the nucleus [53,54]. In the promoter of the gene encoding FKBP51 exists functional GC response elements (GRE), and its activation constitutes an intracellular short feedback loop, which is considered one of the most important hallmarks of GR regulation [55]. FKBP51 expression is induced by GR activation, resulting in a negative feedback loop that regulates GR sensitivity [56]. Moreover, FKBP51 has been described as an important marker for GR sensitivity and bioavailability [11,12]. Interestingly, in adipose tissue biopsies from non-diabetic donors, one of the most responsive genes to dexamethasone is FKBP51 [57]. The higher FKBP51 expression was associated with reduced insulin effects on glucose uptake, suggesting that FKBP51 may be implicated in GC-induced insulin resistance [57]. Moreover, *FKBP5*-knockout mice fed with an HFD are resistant to weight gain and hepatic steatosis, and have reduced adipose tissue [58]. Moreover, *Fkbp5*-knockout mice are protected from HFD-induced weight gain, showing improved glucose tolerance and increased insulin signaling in skeletal muscle through a reduced bind of the PH domain leucine-rich repeat protein phosphatase (PHLPP) with Akt and AS160 [59].

Regular physical activity is one of the most effective non-pharmacological treatments in the therapy of metabolic disorders associated with obesity and its complications [60]. On the other hand, GCs are essential hormones for the exercise adaptations [61] and play an ergogenic role [62]. With this in mind, we evaluated the effect of moderate aerobic training in GC response in CD- and HFD-fed mice. Our results showed that exercise did not produce a body weight loss in the HFD group but improved fasting glucose and tolerance. These results agree with previous reports that showed no effect of exercise upon body weight, but did show metabolic improvements [63,64].

Analysis of blood corticosterone levels indicated that mice subjected to the exercise protocol had higher corticosterone levels than animals without exercise. It is known that exercise stimulates the HPA axis, which results in the release of GCs [65]. By comparing different studies, some authors speculate a positive correlation between the amount of exercise and blood corticosterone levels [66,67]. It should be mentioned that, as in the previous case, corticosterone levels tend to decrease in response to dexamethasone in the HFD group. However, this reduction was not observed in the CD group and could account for exercise-induced resistance to GCs.

Two studies report the role of exercise in models of HFD plus GCs. The first described that 4 weeks of exercise training or the exercise mimetic AICAR improved glucose homeostasis and depressive behavior in a model of depression-like and insulin resistance states by co-treatment with HFD and corticosterone [68]. The second study showed that 2.5 weeks of voluntary exercise attenuated visceral adiposity gain, glucose intolerance, and insulin resistance caused by HFD plus corticosterone [69]. However, the exercise effect on GC sensitivity is still unknown. Our results showed that exercise counteracts the HFD-induced hepatic GR hypersensitivity. In humans, acute and chronic exposure to exercise triggers resistance to GCs in monocytes, inducing a decrease in the inhibition of LPS-stimulated IL-6 release [70,71]. These results suggest that the benefits of exercise could be related to GR resistance. By contrast, the adipose tissue increases its response to GCs, which could be related to an increase in the vascularization of inter-abdominal adipose tissue induced by exercise [72]. In this regard, FKBP51 is the most responsive gene in isolated human omental adipocytes treated with dexamethasone, with its expression correlating negatively with fasting glucose [57,73]. However, these data are in isolated adipocytes. Thus, it is relevant to study whether exercise is able to increase the delivery of dexamethasone to adipose tissue.

In conclusion, we reported that HFD-induced obesity produces a hepatic GR hypersensitivity in male mice. Moreover, the exercise reversed this hypersensitivity in the liver and increased the sensitivity in adipose tissue. Taking account that the deregulation of GC homeostasis is associated with the development of obesity and the alteration of glucose and lipid metabolism, our results suggest that hepatic GR sensitivity may play an important role in development or treatment in the metabolic disorders associated with obesity. However, this is a first characterization of the effects of HFD and exercise in the GR response in key metabolic tissues. A more in-depth investigation is necessary to unravel the metabolic effects of the hepatic GR hypersensitivity.

## 4. Materials and Methods

### 4.1. Animals and Study Design

*Protocol 1.* For this study, we used male C57BL/6J wild type male mice, which were kept under normal humidity (40–60%) and temperature (22–24 °C) in a 12:12 h light–dark cycle. The animals had free access to purified water. After weaning, we treated them for 3 months with 1 of these 2 diets: CD (control diet: 10% protein, 20% carbohydrates, and 70% fat, D12450J, Research Diets) and HFD (high-fat diet: 60% fat, 20% protein, and 20% carbohydrates, D12492, Research Diets). At the end of the treatment, 8 h before euthanasia, we divided the animals into 2 groups: the first group was injected with a single dose of dexamethasone (2 mg/Kg of body weight; D2915, Sigma, St. Louis, MO, USA) intraperitoneally (i.p.), and the second group was injected with phosphate buffered-saline (PBS) i.p.

*Protocol 2.* These animals were housed, fed, divided, and injected under conditions identical to protocol 1, except for the exercise protocol. The training was performed for 12 weeks. The mice were familiar with the treadmill running for 3 days with 1 rest day before starting the protocol. The training began 1 week after weaning along with the diet. There were 3 sessions per week: at the start with 5-min warm-up at 5 m/min, followed by 30 min of training at a speed of 8–12 m/min.

Animals were euthanized by inhalation of isoflurane, and blood and tissues were collected. Animal care and procedures were approved by the Animal Ethics Committee of the University of Chile (17073-INT-UCH; 26-09-17). Male C57BL/6J mice were obtained from the Institute of Public Health (Santiago, Chile) at 3 weeks of age.

### 4.2. Glucose Tolerance Test (GTT)

The mice were separated in individual cages, weighed, and fasted for 6 h before starting the test. The animals were then injected with a sterile glucose solution of 1.5 mg/g body weight, i.p. To evaluate the blood glucose levels, we collected the tail blood immediately before the glucose injection (0) and 15, 30, 90, and 120 min later. Glucose concentration was determined with an Accu-Check Performa glucometer (Roche Diagnostic, Mannheim, Germany).

### 4.3. Corticosterone

The blood was extracted from the animals through the posterior vein cava with previously filled syringes with ethylenediaminetetraacetic acid (EDTA). All blood samples were obtained at 15:00 h. Blood samples were placed in red-capped clinical tubes that were allowed to stand for 60 min in the cold, which were then centrifuged at 2000× *g* for 10 min. Next, plasmas were extracted and deposited in cryotubes, being stored at −80 °C until use. The plasma corticosterone concentration was determined through a commercial enzyme immunoassay kit (Cayman Chemical, Ann Arbor, MI, USA).

### 4.4. RNA Extraction, cDNA Synthesis, and Real-Time PCR

The total RNA from liver and gastrocnemius samples were extracted using the commercial extraction kit (Bio-Rad, USA), according to the manufacturer’s instructions, and frozen at −80 °C until the complementary DNA synthesis. The RNA integrity of each sample was evaluated by the absorbance ratio at 260/280. RNA extraction from adipose tissue samples was performed using a specific adipose tissue commercial kit RNeasy Lipid Tissue Mini Kit (Qiagen, Hilden, Germany) and used according to the manufacturer’s instructions. Total RNA (1 µg) was used for cDNA synthesis with the RevertAid First Strand cDNA Synthesis Kit (Thermo Scientific, Waltham, MA, USA), following the manufacturer’s instructions. For quantifying the mRNA of FKBP51, PDK4, KLF15, 11β-HSD1, and GRα, we used the Eco qPCR system (Illumina, San Diego, CA, USA). The results were analyzed with Eco Real-Time PCR software v4.1 (Illumina). The results were expressed by the geometric mean of 3 housekeeping genes as intrinsic control, and expression of particular conditions was correlated with the control condition of each tissue (PBS-injected). For the gastrocnemius, we used receptor like protein 5 (RLP5), tyrosine 3-monooxygenase/tryptophan 5-monooxygenase activation protein zeta (YWHAZ), and β-actin. For the liver and adipose tissue, we used glyceraldehyde-3-phosphate dehydrogenase (GAPDH), YWHAZ, and β-actin. The primers′ sequences are presented in Appendix A.

### 4.5. Western Blot

Tissue proteins were extracted using 300 µL of radioimmunoprecipitation assay (RIPA) buffer (25 mM Tris-HCl (pH 7.6), 150 mM NaCl, 1.0% sodium deoxycholate, 0.1% SDS) in the presence of a cocktail of protease inhibitor (Roche Diagnostics, Basel, Switzerland)) and phosphatases (Roche Diagnostics, Basel, Switzerland)) and frozen at −80 °C. Protein concentration was determined using BCA Protein assay kit Protein (Sigma, St. Louis, MO, USA) with a standard curve of bovine serum albumin. A total of 30 µg of each sample was separated by electrophoresis in polyacrylamide gels (Tris-glycine SDS-polyacrylamide 10%) and electrotransferred to polyvinylidene fluoride (PVDF) membranes (0.45 µm). The primary antibodies and their dilution were FKBP5 1:1000 (12210S, Cell Signaling), G6Pase 1:500 (83690, Abcam, Cambridge, United Kingdom), and GAPDH 1:5000 (MAB374, Sigma). GAPDH was used as a loading control. The images were scanned using C-Digit and quantified using Image Studio Digits (LICOR Bioscience, Lincon, NE, USA).

### 4.6. Statistical Analysis

All results were represented as the mean ± standard error of mean. Student’s *t*-test of independent samples analyzed the results for the comparisons of two groups. For comparisons greater than two groups, we used a two-way ANOVA test followed by Tukey’s post-test. Values of *p* < 0.05 were considered significant. The statistical analysis was performed using the Prism 7 software (GraphPad).

## Figures and Tables

**Figure 1 ijms-21-07582-f001:**
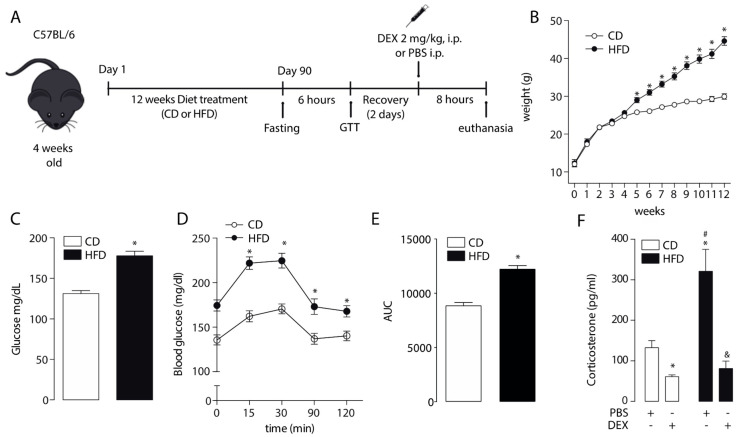
Effect of a high-fat diet on body weight, glucose homeostasis, and corticosterone plasma levels. (**A**) Study design. (**B**) Body weight. (**C**) Fasting glucose. (**D**) Glucose tolerance test (GTT). (**E**) GTT area under the curve (AUC). * *p* < 0.05, unpaired *t*-test. (**F**) Corticosterone plasma levels after a dexamethasone injection (DEX; 2 mg/Kg intraperitoneally (i.p.)) or phosphate-buffered saline (PBS). *n* = 6 mice per condition. Two-way analysis of variance (ANOVA) was conducted, followed by Tukey’s post hoc test. * *p* < 0.05 vs. CD + PBS, ^#^
*p* < 0.05 vs. CD + DEX, ^&^
*p* < 0.05 vs. HFD + PBS. Values are expressed as mean ± standard error of mean.

**Figure 2 ijms-21-07582-f002:**
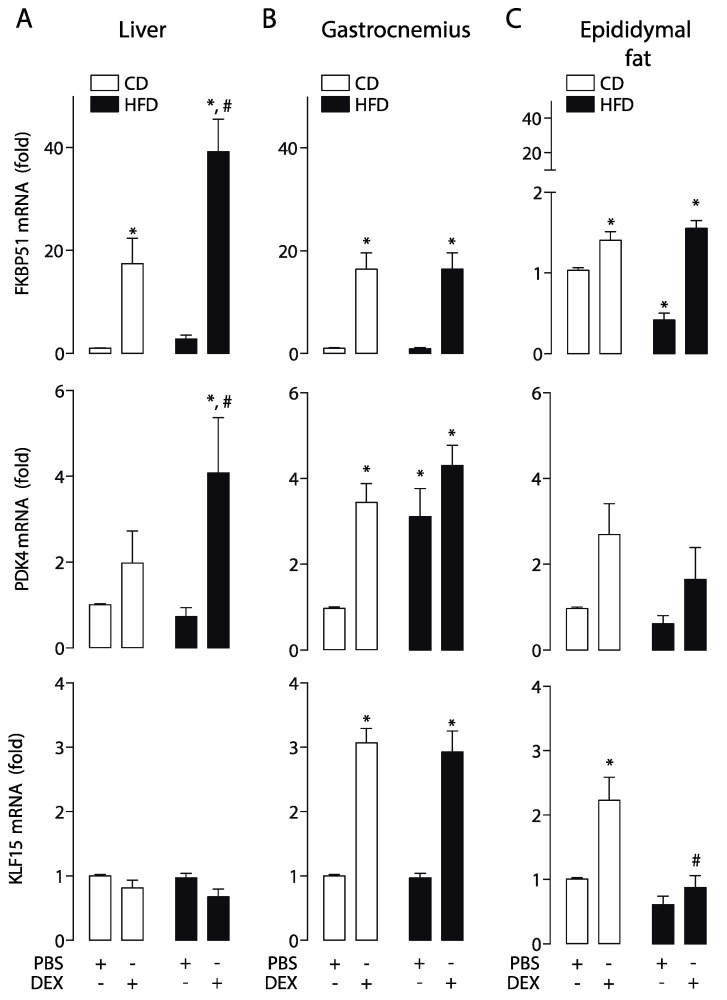
A high-fat diet alters the glucocorticoid receptor (GR) response in a tissue-specific manner. FK506-binding protein 51 (FKBP51), pyruvate dehydrogenase kinase 4 (PDK4), and Krüppel-like factor 15 (KLF15) mRNA levels were determined by qPCR in the liver, gastrocnemius, and epididymal fat obtained from high-fat diet (HFD) or control diet (CD) mice treated with a dexamethasone injection (DEX; 2 mg/Kg i.p.) or phosphate-buffered saline (PBS). (**A**) mRNA expression in the liver. (**B**) mRNA expression in the gastrocnemius. (**C**) mRNA expression in epididymal fat. *n* = 6 mice per condition. Two-way analysis of variance (ANOVA) was conducted, followed by Tukey’s post hoc test. * *p* < 0.05 vs. CD + PBS, ^#^
*p* < 0.05 vs. CD + DEX. Values are expressed as mean ± standard error of mean.

**Figure 3 ijms-21-07582-f003:**
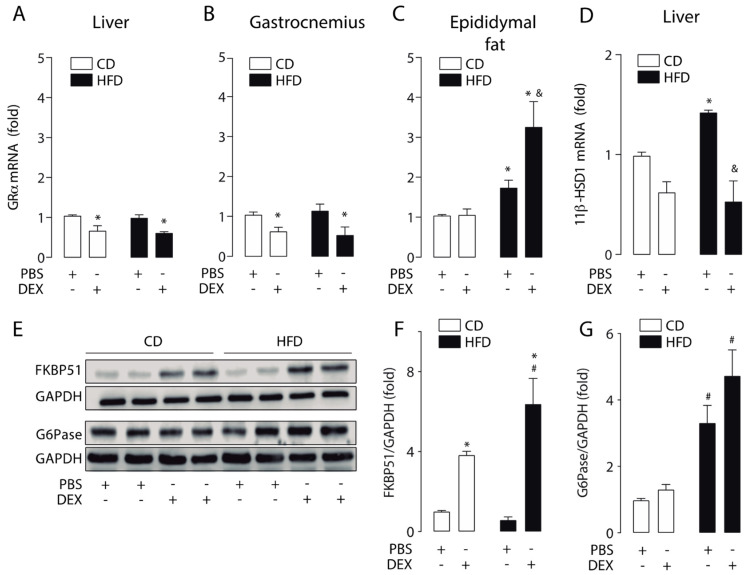
A high-fat diet increased the liver response to glucocorticoids. Glucocorticoid receptor α (GRα) and 11β-hydroxysteroid dehydrogenase type 1 (11β-HSD1) mRNA levels were determined by qPCR in liver, gastrocnemius, and epididymal fat obtained from high-fat diet (HFD) or control diet (CD) mice treated with a dexamethasone injection (DEX; 2 mg/Kg i.p.) or phosphate-buffered saline (PBS). FK506-binding protein 51 (FKBP51) and glucose-6-phosphatase (G6Pase) protein levels were assessed by Western blotting. Glyceraldehyde-3-phosphate dehydrogenase (GAPDH) was used as a loading control. (**A**) GRα mRNA expression in the liver. (**B**) GRα mRNA expression in the gastrocnemius. (**C**) GRα mRNA expression in epididymal fat. (**D**) 11β-HSD1 mRNA expression in the liver. (**E**) Representative Western blot image. (**F**) Liver FKBP51 protein levels. (**G**) Liver G6Pase protein levels. *n* = 6 mice per condition. Two-way analysis of variance (ANOVA) was conducted, followed by Tukey’s post hoc test. * *p* < 0.05 vs. CD + PBS, ^#^
*p* < 0.05 vs. CD + DEX, ^&^
*p* < 0.05 vs. HFD + PBS. Values are expressed as mean ± standard error of mean.

**Figure 4 ijms-21-07582-f004:**
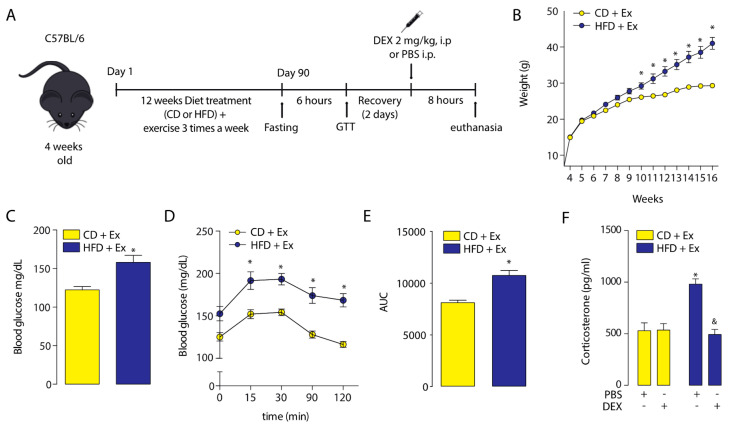
Effects of exercise on body weight, glucose homeostasis, and corticosterone plasma levels in normal and obese mice. (**A**) Study design. (**B**) Body weight. (**C**) Fasting glucose. (**D**) Glucose tolerance test (GTT). (**E**) GTT area under the curve (AUC). * *p* < 0.05, unpaired *t*-test. * *p* < 0.05, unpaired *t*-test. (**F**) Corticosterone plasma levels after a dexamethasone injection (DEX; 2 mg/Kg i.p.) or phosphate-buffered saline (PBS). *n* = 6 mice per condition. Two-way analysis of variance (ANOVA) was conducted, followed by Tukey’s post hoc test. * *p* < 0.05 vs. CD + PBS, ^&^
*p* < 0.05 vs. HFD + PBS. Values are expressed as mean ± standard error of mean.

**Figure 5 ijms-21-07582-f005:**
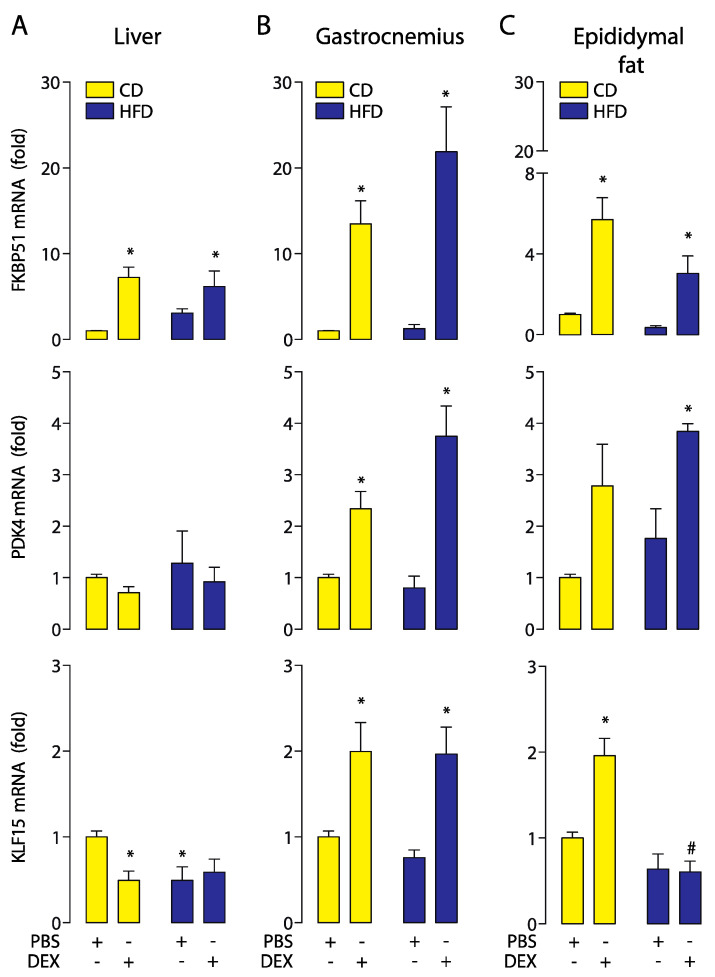
Moderate aerobic training prevented the hepatic GR hypersensitivity. FK506-binding protein 51 (FKBP51), pyruvate dehydrogenase kinase 4 (PDK4), and Krüppel-like factor 15 (KLF15) mRNA levels were determined by qPCR in the liver, gastrocnemius, and epididymal fat obtained from high-fat diet (HFD) or control diet (CD) mice subjected to moderate anaerobic training and treated with a dexamethasone injection (DEX; 2 mg/Kg i.p.) or phosphate-buffered saline (PBS). (**A**) mRNA expression in the liver. (**B**) mRNA expression in the gastrocnemius. (**C**) mRNA expression in epididymal fat. *n* = 6 mice per condition. Two-way analysis of variance (ANOVA) was conducted, followed by Tukey’s post hoc test. * *p* < 0.05 vs. CD + PBS, ^#^
*p* < 0.05 vs. CD + DEX. Values are expressed as mean ± standard error of mean.

**Figure 6 ijms-21-07582-f006:**
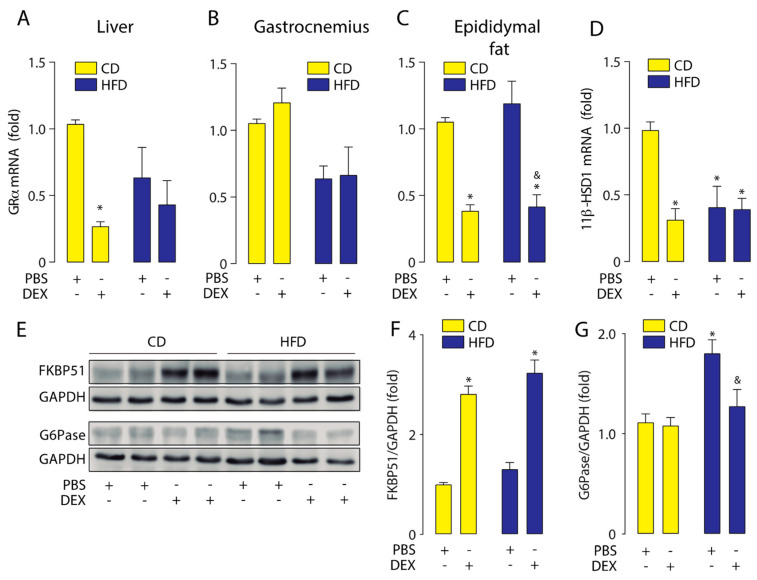
Moderate aerobic training prevented the high-fat diet-increased glucocorticoid receptor response. Glucocorticoid receptor α (GRα) and 11β-hydroxysteroid dehydrogenase type 1 (11β-HSD1) mRNA levels were determined by qPCR in the liver, gastrocnemius, and epididymal fat obtained from high-fat diet (HFD) or control diet (CD) mice subjected to moderate anaerobic training and treated with a dexamethasone injection (DEX; 2 mg/Kg i.p.) or phosphate-buffered saline (PBS). FK506-binding protein 51 (FKBP51) and glucose-6-phosphatase (G6Pase) protein levels were assessed by Western blotting. Glyceraldehyde-3-phosphate dehydrogenase (GAPDH) was used as a loading control. (**A**) GRα mRNA expression in the liver. (**B**) GRα mRNA expression in the gastrocnemius. (**C**) GRα mRNA expression in epididymal fat. (**D**) 11β-HSD1 mRNA expression in the liver. (**E**) Representative Western blot image. (**F**) Liver FKBP51 protein levels. (**G**) Liver G6P protein levels. *n* = 6 mice per condition. Two-way analysis of variance (ANOVA) was conducted, followed by Tukey’s post hoc test. * *p* < 0.05 vs. CD + PBS. ^&^
*p* < 0.05 vs. HFD + PBS. Values are expressed as mean ± standard error of mean.

**Table 1 ijms-21-07582-t001:** Effects of moderate aerobic training in body weight and glucose tolerance in control- and high-fat diet-fed mice.

			Exercise	
	CD	HFD	CD	HFD
**Weight**				
Body (g)	30 ± 0.7	44.6 ± 1.1 *	29.3 ± 0.4	41 ± 1.6 *
Liver (g)	1.46 ± 0.2	1.54 ± 0.2	1.46 + 0.2	1.63 ± 0.2
eWAT (g)	0.72 ± 0.2	2.51 ± 0.4 *	0.71 ± 0.2	2.77 + 0.2 *
**Glucose**				
Fasting blood (mg/dL)	131 ± 3.7	178 ± 5.7 *	122 ± 4.4	158 ± 9.2 ^#^
GTT AUC	9912 ± 295	13,424 ± 315 *	9200 ± 228	12,053 ± 484 *^,#^

Control diet (CD), high-fat diet (HFD), eWAT (epididymal white adipose tissue). Values are expressed as mean ± standard error mean. * *p* < 0.05 vs. CD; ^#^
*p* < 0.05 vs. HFD.

**Table 2 ijms-21-07582-t002:** Effects of moderate aerobic training in dexamethasone-induced gene expression in control- and high-fat diet-fed mice.

			Exercise	
	CD + DEX	HFD + DEX	CD + DEX	HFD + DEX
**Liver**				
FKBP51	17.46 ± 4.9	39.21 ± 6.3 *	7.23 ± 1.2^#^	6.17 ± 1.6 ^#^
PDK4	1.98 ± 0.7	4.08 ± 1.3 *	0.71 + 0.1^#^	0.92 ± 0.3 ^#^
KLF15	0.81 ± 0.1	0.68 ± 0.1	0.49 ± 0.1	0.59 + 0.1
**Gastrocnemius**				
FKBP51	14.07 ± 3.7	13.61 ± 3.5	13.51 ± 4.3	21.9 ± 5.2
PDK4	3.44 ± 0.4	4.3 ± 0.47	2.53 ± 0.4	3.75 ± 0.6
KFLF15	3.07 ± 0.2	2.93 ± 0.3	1.99 ± 0.3	1.97 ± 0.3
**Epididymal fat**				
FKBP51	1.40 ± 0.1	1.55 ± 0.1	5.70 ± 1.1 *	3.05 ± 0.9
PDK4	2.70 ± 0.7	1.05 ± 0.5	2.78 ± 0.8	3.84 ± 0.2 ^#^
KFLF15	2.23 ± 0.4	0.87 ± 0.2 *	1.96 ± 0.2	0.60 ± 0.1 *

Control diet (CD), high-fat diet (HFD). Values are expressed as mean ± SEM. * *p* < 0.05 vs. CD + DEX; ^#^
*p* < 0.05 vs. HFD + DEX.

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
