# Peer review of "Moderate Aerobic Exercise Training Prevents the Augmented Hepatic Glucocorticoid Response Induced by High-Fat Diet in Mice"

_ijms, 2020, doi:10.3390/ijms21207582_

Round 1

Reviewer 1 Report

Comments for authors.

The authors of the manuscript entitled “Moderate aerobic exercise training prevents the augmented hepatic glucocorticoid response induced by high-fat diet in mice” aimed to investigate the effect of HFD-induced obesity in the GCs response in the liver, skeletal muscle, and adipose tissue. Their results showed that HFD fed mice developed a hepatic sensitivity to GCs. This alteration in GC response is prevented by moderate aerobic training. The results found suggest that mice fed with HFD-induced obesity develop a hepatic GR sensitivity, which could enhance the metabolic effects of HFD in the liver. Also, the authors conclude that exercise is a feasible non- pharmacological strategy to prevent the deregulation in GR response in obesity. The present study is very interesting, but there are some points that need to be cleared.

The authors stated in the results section that they used several genes expression to indicate GR sensitivity in HFD-obese control and trained rats. However, this information is not clear in the introduction.

The hypothesis of this study is not clear in the introduction.

Authors should discuss the different responses they found among tissues and what does it means. I understand that it is just a characterization manuscript and more studies need to be done, but it would be great to see a possible mechanistic explanation about the effects of DEX, HFD and exercise among different tissues. It seems that the authors focused only on hepatic GR sensitivity.

Methods: training program needs a reference. Why did the authors choose a 3 days/week exercise program, since most of the aerobic training programs on a treadmill use 5 days/week.

Some verbs in the results session should be in the past tense. Please, check grammar adequately.

I believe that results should be presented together in all figures (like it is shown on table 1), so it would be possible to see some interaction between HFD and exercise

Author Response

All the authors are deeply grateful for the comments and critical points raised by the reviewer because they contributed to improving the scientific contents of our study.

Reviewer’ comments:

Reviewer 1

The authors of the manuscript entitled “Moderate aerobic exercise training prevents the augmented hepatic glucocorticoid response induced by high-fat diet in mice” aimed to investigate the effect of HFD-induced obesity in the GCs response in the liver, skeletal muscle, and adipose tissue. Their results showed that HFD fed mice developed a hepatic sensitivity to GCs. This alteration in GC response is prevented by moderate aerobic training. The results found suggest that mice fed with HFD-induced obesity develop a hepatic GR sensitivity, which could enhance the metabolic effects of HFD in the liver. Also, the authors conclude that exercise is a feasible non- pharmacological strategy to prevent the deregulation in GR response in obesity. The present study is very interesting, but there are some points that need to be cleared.

Query 1:  The authors stated in the results section that they used several genes expression to indicate GR sensitivity in HFD-obese control and trained rats. However, this information is not clear in the introduction.

Response: Thanks for your comment. We added two paragraphs regarding gene expression regulation by glucocorticoids and a description of the target gene FKBP51 (refers lines 47-50 and 53-60).

Query 2: The hypothesis of this study is not clear in the introduction.

Response: We added the hypothesis in the introduction section as follows “Our hypothesis states that high-fat diet-induced obesity deregulated the GCs sensitivity, which is prevented by moderate aerobic exercise training” (refers lines 86-87).

Query 3: Authors should discuss the different responses they found among tissues and what does it means. I understand that it is just a characterization manuscript and more studies need to be done, but it would be great to see a possible mechanistic explanation about the effects of DEX, HFD and exercise among different tissues. It seems that the authors focused only on hepatic GR sensitivity.

Response: We focused mainly on hepatic response due that the principal effects of HFD and exercise were observed in this tissue. The incorporation of the new table 2 clarifies this point when comparing the impact of diet and exercise in GCs response. (refers to table 2, lines 199-206). We also added a brief discussion about gastrocnemius and epididymal fat (refers lines 273-277, 324-327).

Query 4: Methods: training program needs a reference. Why did the authors choose a 3 days/week exercise program, since most of the aerobic training programs on a treadmill use 5 days/week?          

Response: We choose a 3 days/week instead 5 days/week based on non-published data from our lab, where we observed that our exercise protocol improves glucose tolerance without changes in body weight. By contrast, the exercise program of 5 days/week induces a decrease in body weight that could explain in part the effects in GC response. Thus, we can say that the effects seen in the GCs responses are due to exercise and not to an exercise-induced weight loss. Papers demonstrating weight loss with a 5 day/week exercise program are shown.

  • de Silva, L. L. S., de Sousa Fernandes, M. S., Kubrusly, M. S., Muller, C. R., Américo, A. L. V., Stefano, J. T., ... & Jukemura, J. (2020). Effects of Aerobic Exercise Protocol on Genes Related to Insulin Resistance and Inflammation in the Pancreas of ob/ob Mice with NAFLD. Clinical and Experimental Gastroenterology, 13, 223.
  • Stoyell-Conti, F. F., Irigoyen, M. C. C., Sartori, M., Ribeiro, A. A., Santos, F., Machi, J. F., ... & De Angelis, K. (2019). Aerobic training is better than resistance training on cardiac function and autonomic modulation in ob/ob mice. Frontiers in physiology, 10, 1464.
  • Ghareghani, P., Shanaki, M., Ahmadi, S., Khoshdel, A. R., Rezvan, N., Meshkani, R., ... & Gorgani-Firuzjaee, S. (2018). Aerobic endurance training improves nonalcoholic fatty liver disease (NAFLD) features via miR-33 dependent autophagy induction in high fat diet fed mice. Obesity research & clinical practice, 12(1), 80-89.

Query 5: Some verbs in the results session should be in the past tense. Please, check grammar adequately.

Response: Fixed.

Query 6: I believe that results should be presented together in all figures (like it is shown on table 1), so it would be possible to see some interaction between HFD and exercise

Response: Thank you for your valuable comment, in the revised version of the manuscript and in line with the objective of this work, which is to study the GCs sensitivity. We added a new table (table 2) comparing mRNA levels of FKBP51, PDK4, and KLF15 in the liver, gastrocnemius, and epididymal fat in dexamethasone injected mice. With this new table, we can compare the interaction between diet and exercise in the GCs response. (refers to table 2, lines 199-206).

Reviewer 2 Report

Manuscript  by  Dassonvalle j et al.  “  Moderate aerobic exercise training prevents the 2 augmented hepatic glucocorticoid response induced 3 by high-fat diet in mice”  gives interesting data about

consequences of HFD-induced obesity in the GR sensitivity in energetic  homeostasis  relevant tissues and the effects of moderate aerobic training in this response.

Authors are presenting data form 2 mouse models involving 4 animal groups and testing different parameters related to their question.  Manuscript needs major  improvement in concern with data analysis (details under 8.Line 169…)

 Manuscript need for some English improvements and further issues are listed:

  1. In introduction give few sentences about genes /proteins you have investigated  to show why are they used in your investigation.. (FKBP51), pyruvate dehydrogenase kinase 4 (PDK4), and Krüppel-like factor 15 (KLF15), G6Pase

2.Line 77-709: please rewrite the sentence in active form …I this work we have investigated….

3.Line 82, line 308  : naming mouse strain  with “mice C57BL6”  is not informative at all since there are many sub strains with different genetic mutations involved so please specify exactly the mouse model strain .  

4.Line 87 and figure 1 and figure 4: A) Presently  it seems that all mice had 12 weeks of HFD??? Change that into  12 weeks Diet treatment (CD or HFD. ) Please indicate  in a picture how long is recovery period , and indicate that animals received DEX or PBS

5.Line 90.HDF in HFD

6.Line 128 “that mRNA” into “as mRNA”

7.Line 158,159- table text should be placed directly above table

8.Line 169 :  Chapter:2.3  and  2.4 … major issue : data presented here are comparing 2 diet mice groups  exposed to the same  exercise procedure  (figure 4A) ; so what you have here is diet effect since this is the major difference between the groups.

To see the effect of exercise you have to compare: A) control diet mice without exercise versus control diet mice exposed to  exercise; B)  HFD mice without exercise procedure versus HFD mice exposed to  exercise .  This should be presented in a new would be new Figure and chapter…accordingly discussion needs to be changed.

9.Line 348,349--- data were normalized with  3 referent genes as intrinsic control and expression of particular condition was correlated to control condition of each organ (PBS treated animals: this is set to 1 on all of your Figures) ..

10.Nomenclature RT-qPCR is used for direct detection from mRNA and you have made as first step RT-PCR and used total cDNA for detection of specific gene – this should be qPCR) …please check the nomenclature RT- qPCR   

Author Response

All the authors are deeply grateful for the comments and critical points raised by the reviewer because they contributed to improving the scientific contents of our study.

Reviewer’ comments:

Reviewer 2

Manuscript by Dassonvalle j et al.  “Moderate aerobic exercise training prevents the 2 augmented hepatic glucocorticoid response induced 3 by high-fat diet in mice” gives interesting data about consequences of HFD-induced obesity in the GR sensitivity in energetic homeostasis relevant tissues and the effects of moderate aerobic training in this response.

Authors are presenting data form 2 mouse models involving 4 animal groups and testing different parameters related to their question.  Manuscript needs major improvement in concern with data analysis (details under 8. Line 169…)

 Manuscript need for some English improvements and further issues are listed:

Query 1: In introduction give few sentences about genes /proteins you have investigated to show why they are used in your investigation. (FKBP51), pyruvate dehydrogenase kinase 4 (PDK4), and Krüppel-like factor 15 (KLF15), G6Pase.

Response: Thanks for your comment. We added to the introduction section information about the GCs target genes used in our investigation.  (refers lines 47-50, 52-60)

Query 2: Line 77-709: please rewrite the sentence in active form …I this work we have investigated….

 Response: Fixed.

Query 3: Line 82, line 308: naming mouse strain with “mice C57BL6” is not informative at all since there are many sub strains with different genetic mutations involved so please specify exactly the mouse model strain.   

Response: Thank you for this valuable correction. The mice strain is C57BL/6J. We added this information in the manuscript.

Query 4: Line 87 and figure 1 and figure 4: A) presently it seems that all mice had 12 weeks of HFD??? Change that into 12 weeks Diet treatment (CD or HFD) Please indicate in a picture how long is recovery period, and indicate that animals received DEX or PBS

Response: According to the reviewer's suggestions, we fixed the figures.

Query 5: Line 90.HDF in HFD

Response: Fixed.

Query 6: Line 128 “that mRNA” into “as mRNA”

Response: Fixed.

Query 7: Line 158,159- table text should be placed directly above table

Response: Fixed

Query 8: Line 169:  Chapter:2.3 and 2.4 … major issue: data presented here are comparing 2 diet mice groups exposed to the same exercise procedure (figure 4A); so, what you have here is diet effect since this is the major difference between the groups.

Response: Thank you for your valuable comment, in the revised version of the manuscript and in line with the objective of this work, which is to study the GCs sensitivity. We added a new table (table 2) comparing mRNA levels of FKBP51, PDK4, and KLF15 in the liver, gastrocnemius, and epididymal fat in mice injected with dexamethasone. With this new table, we can compare the interaction between diet and exercise in the GCs response. (refers to table 2, lines 199-206).

Query 9: To see the effect of exercise you have to compare: A) control diet mice without exercise versus control diet mice exposed to exercise; B) HFD mice without exercise procedure versus HFD mice exposed to exercise.  This should be presented in a new would be new Figure and chapter…accordingly discussion needs to be changed.

Response: The new table 2, resume the main effects of GCs sensitivity in the different tissues and the interaction between diet and exercise. This table shows the main effects of exercise are in the FKBP51 and PDK4 mRNA levels in the HFD group. In view of these results, we think that the addition of this new table allows us to understand the effect of exercise in GC sensitivity independent of diet. (refers to table 2, lines 199-206).

Query 10: Line 348,349--- data were normalized with 3 referent genes as intrinsic control and expression of particular condition was correlated to control condition of each organ (PBS treated animals: this is set to 1 on all of your Figures).

Response: Fixed. (refers lines 379-380)

Query 11: Nomenclature RT-qPCR is used for direct detection from mRNA and you have made as first step RT-PCR and used total cDNA for detection of specific gene – this should be qPCR) …please check the nomenclature RT- qPCR

Response: According to the suggestions of the reviewer, we correct it to qPCR.

Round 2

Reviewer 1 Report

The authors responded adequately to all my concerns.